# A prosthesis utilizing natural vestibular encoding strategies improves sensorimotor performance in monkeys

**Kantapon Pum Wiboonsaksakul** [1,2], **Dale C. Roberts** [1], **Charles C. Della Santina** [3], **Kathleen E. Cullen** [1,2,3,4] *

**1** Department of Biomedical Engineering, Johns Hopkins University School of Medicine; Baltimore, Maryland, United States of America, **2** Kavli Neuroscience Discovery Institute, Johns Hopkins University; Baltimore, Maryland, United States of America, **3** Department of Otolaryngology-Head and Neck Surgery, Johns Hopkins University School of Medicine; Baltimore, Maryland, United States of America, **4** Department of Neuroscience, Johns Hopkins University School of Medicine; Baltimore, Maryland, United States of America

* Kathleen.Cullen@jhu.edu

**Data Availability Statement:** All data associated with this study are present in the paper or the

## Abstract

Sensory pathways provide complex and multifaceted information to the brain. Recent advances have created new opportunities for applying our understanding of the brain to sensory prothesis development. Yet complex sensor physiology, limited numbers of electrodes, and nonspecific stimulation have proven to be a challenge for many sensory systems. In contrast, the vestibular system is uniquely suited for prosthesis development. Its peripheral anatomy allows site-specific stimulation of 3 separate sensory organs that encode distinct directions of head motion. Accordingly, here, we investigated whether implementing natural encoding strategies improves vestibular prosthesis performance. The eye movements produced by the vestibulo-ocular reflex (VOR), which plays an essential role in maintaining visual stability, were measured to quantify performance. Overall, implementing the natural tuning dynamics of vestibular afferents produced more temporally accurate VOR eye movements. Exploration of the parameter space further revealed that more dynamic tunings were not beneficial due to saturation and unnatural phase advances. Trends were comparable for stimulation encoding virtual versus physical head rotations, with gains enhanced in the latter case. Finally, using computational methods, we found that the same simple model explained the eye movements evoked by sinusoidal and transient stimulation and that a stimulation efficacy substantially less than 100% could account for our results. Taken together, our results establish that prosthesis encodings that incorporate naturalistic afferent dynamics and account for activation efficacy are well suited for restoration of gaze stability. More generally, these results emphasize the benefits of leveraging the brain's endogenous coding strategies in prosthesis development to improve functional outcomes.

Supplementary Materials. The dataset is available from Zenodo database: https://doi.org/10.5281/zenodo.6338639.

**Funding:** This work was supported by National Institutes of Health grant R01DC002390 (to KEC and CCDS) and grant R01DC018061 (to KEC). The funders had no role in study design, data collection and analysis, decision to publish, or preparation of the manuscript.

**Competing interests:** I have read the journal's policy and the authors of this manuscript have the following competing interests: C.C.D.S. is an inventor of pending and awarded patents related to technologies discussed in this manuscript, and he holds an equity interest in and is the CEO/CSO of Labyrinth Devices, LLC. The terms of this arrangement are managed in accordance with Johns Hopkins University policies on conflict of interest.

## Introduction

Even the simplest sensorimotor transformations require precise neural dynamics. For example, the visuomotor transformations required to produce voluntary saccadic and smooth pursuit eye movements must precisely account for the biomechanical properties of the extraocular muscles and orbital connective tissue (reviewed in [1,2]). Recent developments of sensory prostheses have focused on mimicking the response dynamics of peripheral sensors/afferent systems to improve the functional outcomes of retinal [3] and cochlear [4,5] implants, as well as prostheses aimed at restoring tactile sensation [6–9]. However, complex sensor physiology, limited number of electrodes, and nonspecific stimulation have proven to be a challenge to the development of biomimetic prostheses in these sensory systems.

The vestibular system detects our head motion relative to space. Rotational head motion is sensed by the 3 semicircular canals and linear acceleration is sensed by the 2 otolith organs. In turn, this information is used to generate essential stabilizing reflexes and complex motor synergies that control gaze and posture, in addition to providing us with our subjective sense of motion and orientation. There are many reasons why the vestibular system is uniquely well suited for the development of neural prostheses for the restoration of sensory function. First, the 3 axes of head rotation are encoded by 3 separate semicircular canals, each innervated by a distinct nerve bundle. This sensor structure allows for targeted stimulation of each of the associated orthogonal rotational axes. Second, vestibular afferent response dynamics have been extensively studied and are well understood ([10–13]; also reviewed in [14]). Third, the central pathways mediating essential vestibular reflexes, such as the vestibulo-ocular reflex (VOR), which serves to stabilize gaze during head motion, are simple—comprising 3-neuron circuits in their most direct forms (reviewed in [14]). As a result, the vestibulo-motor transformations generating these rapid stereotyped behavioral responses can be objectively quantified and utilized as a direct measure of prosthesis performance [15–17]. Despite these advantages, whether leveraging the brain's endogenous coding strategies can improve vestibular prosthesis performance surprisingly remains unknown.

To address this question, here, we directly tested whether implementing the natural response dynamics of vestibular afferents improves prosthesis performance in a nonhuman primate model. We integrated these response dynamics into the mapping between head motion and pulse rate delivered by a vestibular prosthesis targeting the ampullary nerve innervating the horizontal semicircular canal [16–19]. Quantification of the resulting eye movements produced by the VOR provided a direct measure of prosthesis performance. Overall, we found that biomimetic mappings that accounted for the brain's endogenous dynamic coding strategies produced more temporally accurate VOR eye movements than those that did not. Further exploration of the parameter space (i.e., the variables that control the mapping of head movement into stimulation rate) revealed that incorporating even more extreme tunings than those naturally displayed by vestibular afferents provided unnatural phase advances, as well as undesirable nonlinear gain saturation. Trends were comparable for stimulation encoding virtual versus physical head rotations, with gains enhanced in the latter case. Using computational methods, we then demonstrated that the same model could explain the eye movements evoked by both sinusoidal and transient stimulation and that a stimulation efficacy substantially less than 100% could account for our results—i.e., each stimulation pulse does not always evoke an afferent action potential. Thus, taken together, our results show that mappings incorporating naturalistic afferent dynamics and stimulation efficacy improve the restoration of gaze stability. More broadly, these results also underscore the benefits of leveraging the brain's natural coding strategies to improve functional outcome of sensory prostheses.

## Results

### Natural encoding strategies improve sensorimotor performance

Vestibular afferents are classified as either regular or irregular based on differences in their resting discharge variability as well as morphology (Fig 1A). While both afferent types show similar response dynamics, namely an increase in gain and phase lead as a function of head motion frequency, irregular afferents exhibit steeper increases of both as a function of frequency, as compared to their regular counterparts (reviewed in [14]). Accordingly, we first implemented 2 biomimetic mappings that accounted for this difference in the natural response dynamics of vestibular afferents, namely a "regular mapping" (Fig 1B, blue) and an "irregular mapping" (Fig 1B, red) in reference to the afferent type that they mimic. We then compared the performance of these 2 biomimetic mappings with that of the conventionally used mapping (e.g., [15,20,21]), which we termed the "static mapping" (Fig 1B, gray). Notably, this latter mapping is characterized by flat gain and no phase lead across the same frequency range, as opposed to the high-pass tuning that is actually demonstrated by vestibular afferents. At low head rotation frequencies, the stimulation rates produced by the 3 mappings were similar, as can be seen in the overlap of the traces in Fig 1C. In contrast, at high frequencies, the stimulation rates of both regular and irregular mappings produced greater depth of modulation and increased phase leads due to their high-pass tuning (Fig 1D).

To establish how implementation of the above mappings impacted performance, we measured and compared the VOR eye movements evoked by the vestibular prosthesis. Two monkeys with total bilateral vestibular loss were implanted with prosthetic electrodes targeted to the ampullary nerve innervating the horizontal semicircular canal. Frequency-modulated pulsatile stimulation was then delivered in darkness to the head-restrained monkeys. These pulses encode virtual head rotations spanning the natural range (0.2 to 20 Hz) (see Materials and methods). Examples of the resultant VOR eye movement evoked in 1 monkey are shown in Fig 2A. To quantify the evoked VOR responses for each of the 3 mappings, we computed the gain and phase at each frequency (Fig 2B and 2C). Gains were normalized to facilitate comparison across animals (see Fig 2B top insets and also S1 Fig for corresponding curves prior to normalization). Notably, while prosthesis stimulation produced comparable VOR gains and phases across all mappings at lower frequencies (0.2 to 1 Hz), this was not the case for stimulation at higher frequencies (e.g., 5 to 20 Hz). Instead, the VOR gain decreased as a function of frequency for the static mapping (negative log-linear slope, linear regression, $p < 0.001$ for both monkeys; Fig 2B, bottom inset), while it increased as a function of frequency for both afferent-like mappings (positive log-linear slope, linear regression, $p < 0.001$ for both monkeys and mappings; Fig 2B, bottom inset). Correspondingly, the VOR phase increasingly lagged the head motion at high frequencies (5 to 20 Hz) for the static mapping while it remained more compensatory for the regular and irregular mappings (i.e., significantly closer to zero degrees, $p < 0.001$ for both monkeys and mappings, Bonferroni corrected). To directly compare the phase compensation across mappings, we then computed mean phase deviations from zero (the mean absolute value) for each mapping over the 5 to 20 Hz stimulation. On average, the static mapping yielded a significantly higher phase deviation than both regular and irregular mappings ($p < 0.001$ for both monkeys and mappings, Bonferroni corrected; Fig 2C, inset). In addition, there was no significant difference in the gain or phase responses of early versus late cycles in either monkey, indicating that no adaption occurred over time ($p > 0.05$, Bonferroni corrected). Taken together, these results show that implementing afferent-like high-pass tuning in the prosthesis mapping overall resulted in more robust high-frequency VOR gains and more compensatory phase behavior.

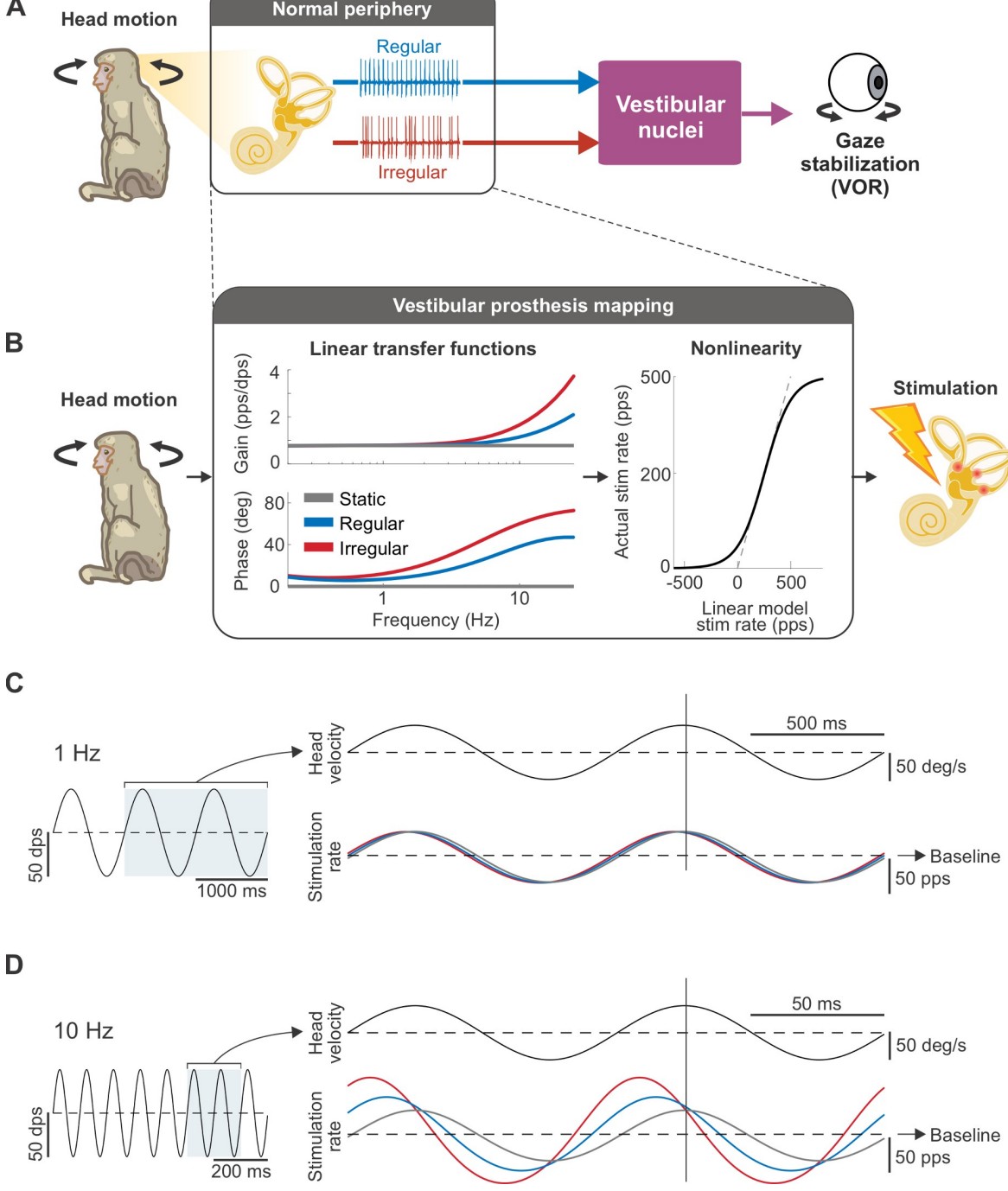

**Fig 1. Biomimetic afferent response dynamics are implemented in the vestibular prosthesis mapping between head motion and stimulation rate.** (**A**) Schematic of the VOR pathway. Both types of vestibular afferents (regular and irregular) convey head movement information to the vestibular nuclei, which, in turn, stabilize the gaze via the VOR. (**B**) Schematic of how the prosthesis converts head movements into pulsatile stimulation. The mapping (black box) consists of the linear transfer functions, which mimic the response dynamics of regular (blue) and irregular (red) afferents or represent the conventional mapping with static gain and phase lead (gray), in cascade with the sigmoidal nonlinearity, which limits the firing rate to be above zero and below the maximum rate of 500 Hz. (**C**) Example of the stimulation rate, modulated around the baseline rate of 150 pps, in response to sinusoidal head movements at 1 Hz. At this frequency, the stimulation rates are similar for all mappings. Vertical line denotes the peak of head movement for phase comparison. (**D**) Example of the stimulation rate in response to sinusoidal head movements at 10 Hz. Both regular and irregular mappings display greater depth of modulation in firing and bigger phase lead than the static mapping, which remains in-phase and shows the same depth of modulation as in (C).

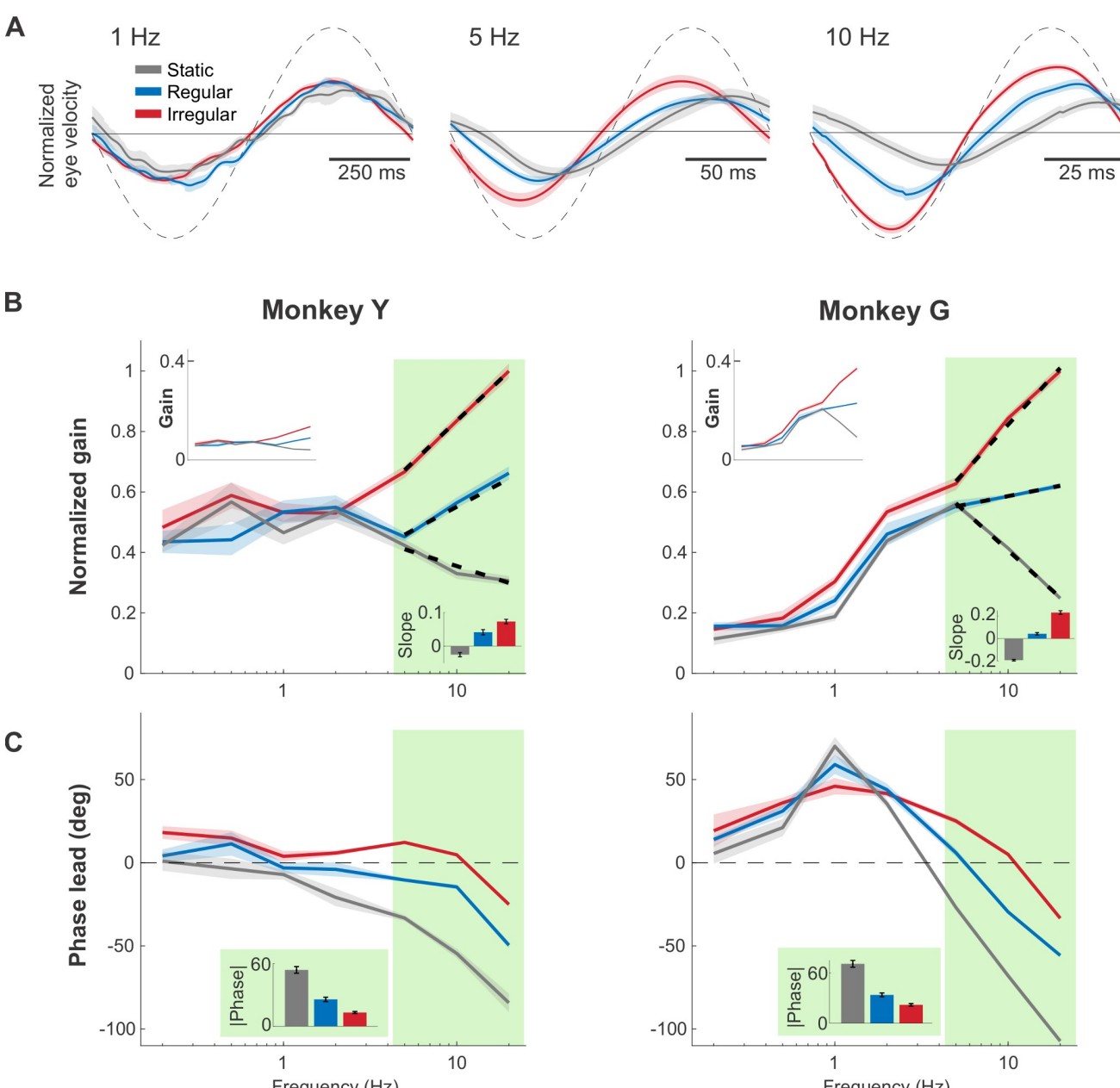

**Fig 2. Afferent-like high-pass dynamics helped maintain robust VOR gains and minimize VOR phase lags at high head rotation frequencies.** (**A**) Example traces of the evoked eye movements during 1, 5, and 10 Hz virtual sinusoidal rotations, normalized to the peak evoked response (i.e., response evoked by the irregular mapping at 20 Hz). Note that the vestibular prosthesis stimulation induced contralaterally versus ipsilaterally directed VOR eye movements in response to increasing versus decreasing stimulation rate relative to the baseline, respectively. Dashed line indicated the simulated head velocity, scaled to facilitate phase response comparison. (**B**) The normalized VOR gains across natural frequency range (0.2–20 Hz), using the same normalization reference as in (**A**). The shaded areas around each trace indicate ± SEM. Light green shading indicates the higher frequencies (5–20 Hz) where the differences between each mapping are more pronounced. The superimposed dashed lines are the log-linear fit over the high frequency range. (Top inset) The VOR gain prior to normalization. (Bottom inset) The log-linear slope of the fit. (**C**) The phase response of the VOR. (Inset) The mean absolute phase at the high frequencies for each mapping. Red, blue, and gray refer to the irregular, regular, and static mappings, respectively. Data underlying this figure can be found at https://doi.org/10.5281/zenodo.6338639.

## Probing parameter space further establishes the advantages of leveraging the brain's natural sensory coding strategy

Our above findings led to the question—would a mapping with even greater high-pass tuning further improve VOR performance? Importantly, testing this hypothesis would be impossible with natural stimulation. However, because we control exactly how the prosthesis encodes head motion, here, we could activate the vestibular afferents in ways natural head motion cannot. To test this possibility, we implemented a novel mapping characterized by a steeper gain and phase lead increase than the natural irregular mapping—which we termed the "super high-pass" mapping (Fig 3A, yellow; see Materials and methods). Examples of the evoked VOR are shown in Fig 3B. Quantification of the gain and phase of the evoked VOR responses are shown in Fig 3C and 3D. Compared to either of our 2 natural (i.e., regular and irregular) afferent mappings, the super high-pass mapping evoked significantly larger VOR responses at high frequencies (Fig 3C; $p < 0.05$ for both monkeys, Bonferroni corrected, 10 to 20 Hz; gains prior to normalization are shown in S2 Fig). Overall, the extreme high-pass nature of this mapping resulted in an approximately 4- and 14-fold increase in gain for 20 versus 0.2 Hz stimulation, for Monkeys Y and G, respectively. Notably, such a steep gain increase as function of frequency differs markedly from the VOR gain in healthy animals, which is flat across the same frequency range [22,23]. Further, the super high-pass mapping overcompensated for the VOR pathway delay, such that the resulting VOR unnaturally led the stimulus at all frequencies except for 20 Hz ($p < 0.01$ for both monkeys, Bonferroni corrected; Fig 3D, yellow).

We then further explored the mapping parameter space to account for the fact that central vestibular neurons that generate the VOR actually receive a mixed input from both afferent types [24–27]. Thus, we tested an additional "mixed" mapping (Fig 3A, purple), with the response dynamics designed to be partway between those of regular and irregular afferents (see Materials and methods). This mapping produced both robust VOR responses and relatively compensatory phase as compared to the static mapping across the tested frequency range (purple curves, Fig 3C and 3D). Specifically, as expected, quantification of the evoked eye movements revealed VOR response dynamics that roughly fell between those of the regular and irregular mappings (compare purple to red and blue curves). Taken together, our results so far show that VOR performance is directly impacted by the level of high-pass tuning in the prosthesis mapping and that biomimetic mappings (regular, mixed, and irregular) yielded better phase compensation than unphysiological mappings (static and super high-pass).

Finally, in healthy animals and human participants, the gain of the VOR is typically close to unity over the frequency range tested here (0.2 to 20 Hz) [22,23]. In contrast, the gain of the responses evoked by our natural afferent mappings above were generally substantially less for both monkeys. Thus, we next asked whether we could obtain higher response gains while maintaining response phase. To do this, we doubled the overall mapping gains without altering the mapping dynamics (see Materials and methods) and analyzed the evoked VOR response (S3 Fig) as described above. Consistent with our prediction, quantification of the gain at each frequency revealed a corresponding increase approaching 2× for the regular, irregular, mixed, and static mappings, except for the super high-pass mapping at 20 Hz, where the response for the doubled gain mapping was significantly lower than 2× in both animals (Fig 3E, top, red arrows; $p < 0.01$ for both monkeys, Bonferroni corrected). This gain decrease is indicative of response saturation—a point which is further discussed below. Additionally, response phases remained mostly unchanged across all frequencies as compared to the original gain condition (compare Figs S3 to 3D). We quantified this observation by calculating the change in phase at each frequency (Fig 3E, bottom). In general, changes in phase were relatively minor with any significant changes ($p < 0.05$, Bonferroni corrected) less than 13 deg. Together, these findings

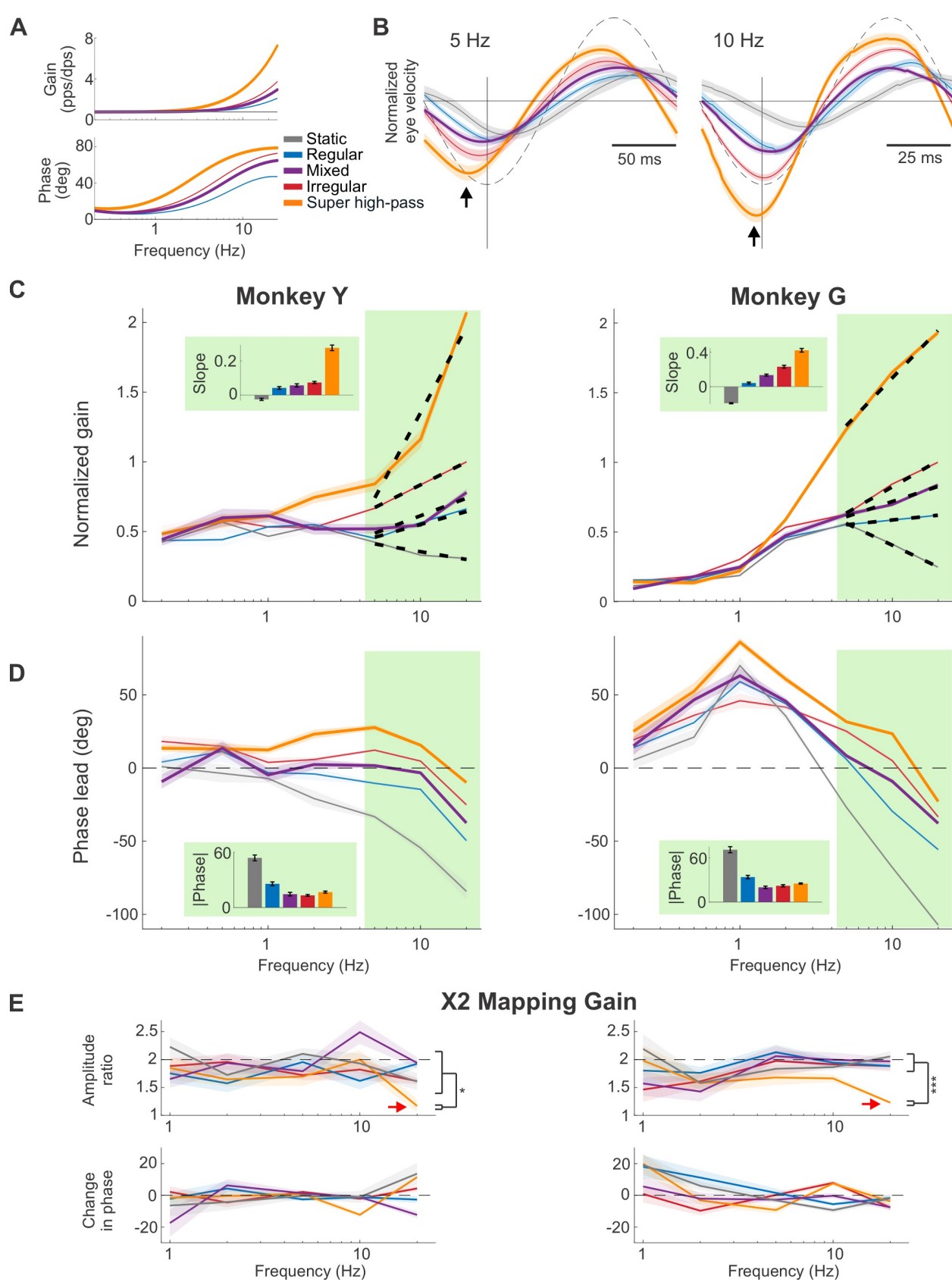

**Fig 3. VOR performance is directly impacted by the response dynamics of the prosthesis mapping and the biomimetic mappings (i.e., regular, mixed, and irregular) resulted in the most accurate timing.** (**A**) The transfer functions of the additional mappings to explore the high-pass tuning parameter space: the super high-pass mapping (yellow), which shows an even higher increase in gain and phase lead than normal irregular afferents do, and the mixed mapping (purple), which was designed to represent the combined responses of regular and irregular afferents. (**B**) Example traces of the normalized evoked eye movements in 1 monkey at 5 and 10 Hz, again normalized to the maximum velocity evoked by the irregular mapping as in Fig 2. Results from Fig 2 are also plotted in thinner line for comparison. Dashed line indicated the simulated head velocity, scaled to facilitate phase response comparison. The vertical black lines indicate the maximum of the virtual head movement stimulus for comparison with the evoked eye velocity. The black arrows denote the peaks of the response from the super high-pass mapping, which showed larger phase lead than the other mappings. (**C** and **D**) Similar plots to (B) and (C) of Fig 2, respectively, with the addition of the data from the 2 new mappings. (**E**) Quantification of the VOR responses after doubling the overall mapping gains (i.e., 2× mapping gains; see Materials and methods). Top row shows the amplitude ratio of the evoked eye velocity of Monkey Y (left) and Monkey G (right) after doubling mapping gains for each of our 5 conditions (i.e., the velocity evoked by the 2× mapping/the velocity evoked by the original 1× mapping). The red arrows point to the ratio from super high-pass mapping at 20 Hz, which is significantly lower than other mappings and indicative of response saturation. Bottom row shows the change in phase after doubling the overall mapping gains. For both monkeys, the changes are generally not significant, with some significant but small (<13 deg) changes ($p < 0.05$, Bonferroni corrected). Data underlying this figure can be found at https://doi.org/10.5281/zenodo.6338639.

establish that we could, in general, obtain higher VOR gains by increasing the overall mapping gains without altering the VOR phase response.

So far, we have focused on the prosthesis-evoked VOR eye movements during virtual head rotations in darkness to isolate the contribution of the vestibular system. We next investigated whether VOR gain was enhanced when prosthetic stimulation patterns were evoked by actual physical motion rather than virtual head rotations (see Materials and methods). Overall, we observed similar trends for the physical and virtual stimulation conditions (compare Figs S4 and 3) in darkness across mappings. We also investigated whether VOR gain was enhanced when prosthetic stimulation patterns were evoked by physical rotations in the light (see Materials and methods) and again found similar trends (compare Figs S5 and 3). Interestingly, higher gains were observed in the physical rotation conditions versus virtual rotation condition mainly for frequencies >5 Hz range for Monkey G ($p < 0.05$, Bonferroni corrected, in darkness and light) and at 10 Hz for Monkey Y in light ($p < 0.05$, Bonferroni corrected). Thus, together these results indicate that biomimetic mappings likewise improved VOR performance during physical head rotations, while also showing gain enhancement at high frequencies. We further consider the implications of this finding in the discussion.

## Improvements in performance generalize to more behaviorally relevant sensory stimuli

In everyday life, we generate transient gaze-orienting head movements that are more complex than a single frequency [28,29]. Accordingly, we next analyzed the VOR evoked by stimulation encoding a head movement representative of monkey's orienting gaze behavior (i.e., [30]; Fig 4A, dashed lines). First, we compared the amplitude of the VOR evoked across mappings. As expected from our above findings using sinusoidal stimulation, VOR gains increased systematically for mappings with greater and greater high-pass tuning (Fig 4B). Next, to quantify VOR timing accuracy, we estimated the time difference between the peak of the encoded head velocity and that of the evoked eye velocity. The biomimetic regular and mixed mappings evoked a peak VOR response that was well aligned with peak head velocity. (Fig 4C, no significant lead for Monkey Y, and only a small but significant lead of 4.1 ± 0.8 ms in Monkey G for the mixed mapping, $p < 0.01$, Bonferroni corrected). In contrast, the VOR evoked by the static mapping was significantly delayed relative to the head velocity profile ($p < 0.01$, Bonferroni corrected, lags of 12 ± 0.7 ms and 12 ± 1.4 for Monkeys G and Y, respectively), while that evoked by irregular and super high-pass mappings actually led the stimulus ($p < 0.05$, Bonferroni corrected, Monkey G: lead of 7.1 ± 0.8 ms and 14.3 ± 0.9 ms for irregular and super-irregular mappings, respectively, Monkey Y: lead of 16.9 ± 1 ms for super-irregular mapping, no significance for

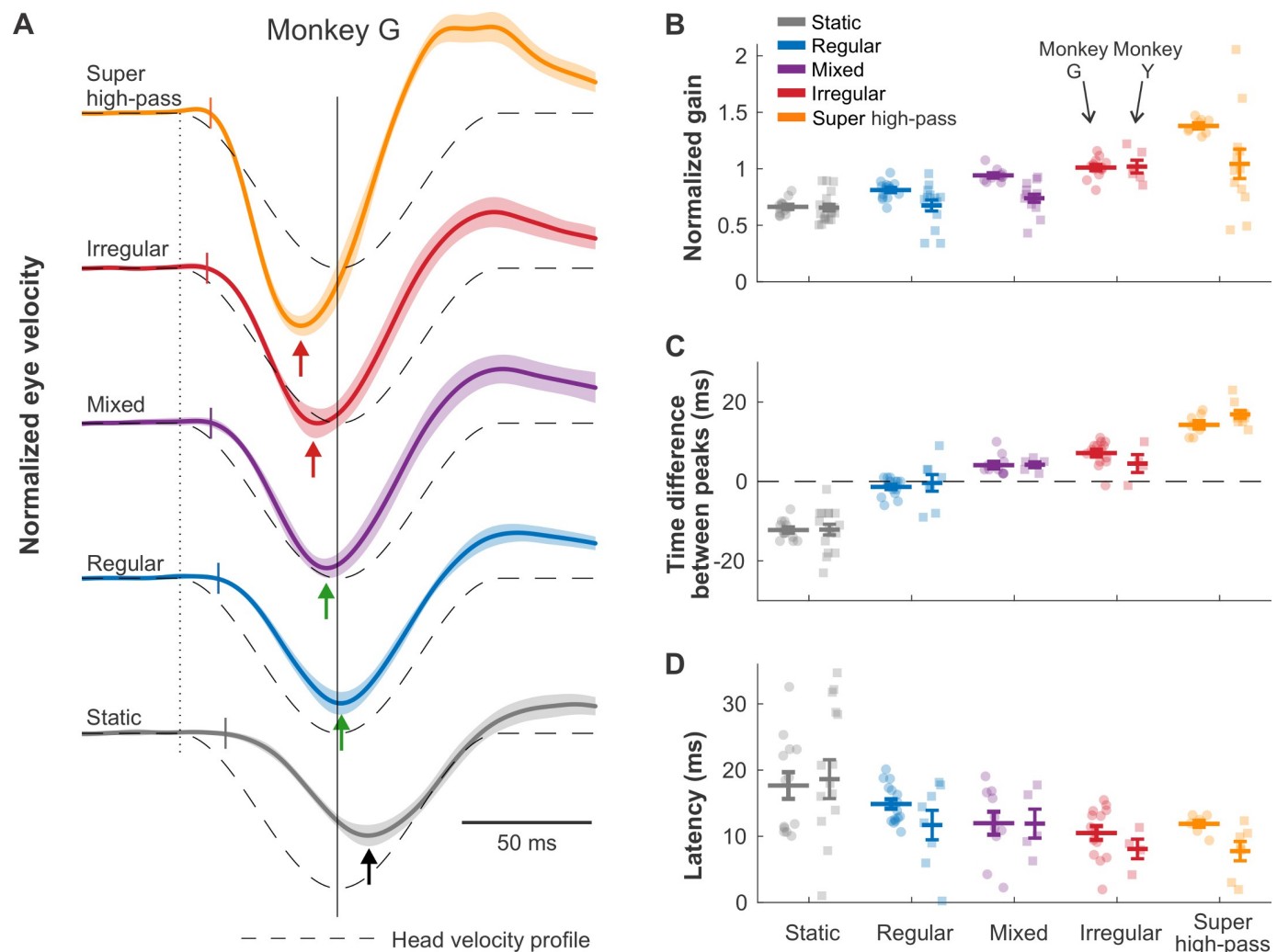

**Fig 4. Afferent-like response dynamics improve timing accuracy of transient VOR.** (**A**) Traces of the evoked eye movements for Monkey G during virtual transient head movements (on direction), again normalized to the maximum velocity evoked by the irregular mapping as in Figs 2 and 3. Dashed lines indicate the inverted head velocity. Dotted vertical line indicates the start of the head movement. Solid vertical line denotes the peak of the inverted head movement. Short colored vertical lines indicated the estimated latency. Arrows show the peak of the eye movement response. (**B**, **C**, and **D**) Quantification of the traces in A for normalized gain [using the same normalization reference as in (**A**)], time difference between eye and head velocity peaks, and onset latency, respectively. Results from Monkey G and Monkey Y are plotted on the left and on the right, respectively. Error bars indicate the SEM. Data underlying this figure can be found at https://doi.org/10.5281/zenodo.6338639.

irregular mapping). Finally, we computed the onset latencies of the VOR evoked across mappings and found that, while the latencies were shortest (approximately 7 to 11 ms) for our 2 highest-pass mappings, this difference did not reach significance (Fig 4D). Comparable results were also found for off-direction transient stimulation (S6 Fig). Finally, for completeness, we compared responses evoked by stimulation encoding physical versus virtual motion (compare Figs S7 to 4). As was shown above for sinusoidal stimuli, similar trends were observed across mappings, and, in general, gains were significantly enhanced for physical motion in both animals ($p < 0.05$, Bonferroni corrected).

Overall, our results using both sinusoidal and transient stimuli revealed parallel trends. Thus, we next asked whether the same model could account for the eye movements evoked by both types of stimulation. To test this proposal, we first fit a simple linear model (Fig 5A; see

**A**

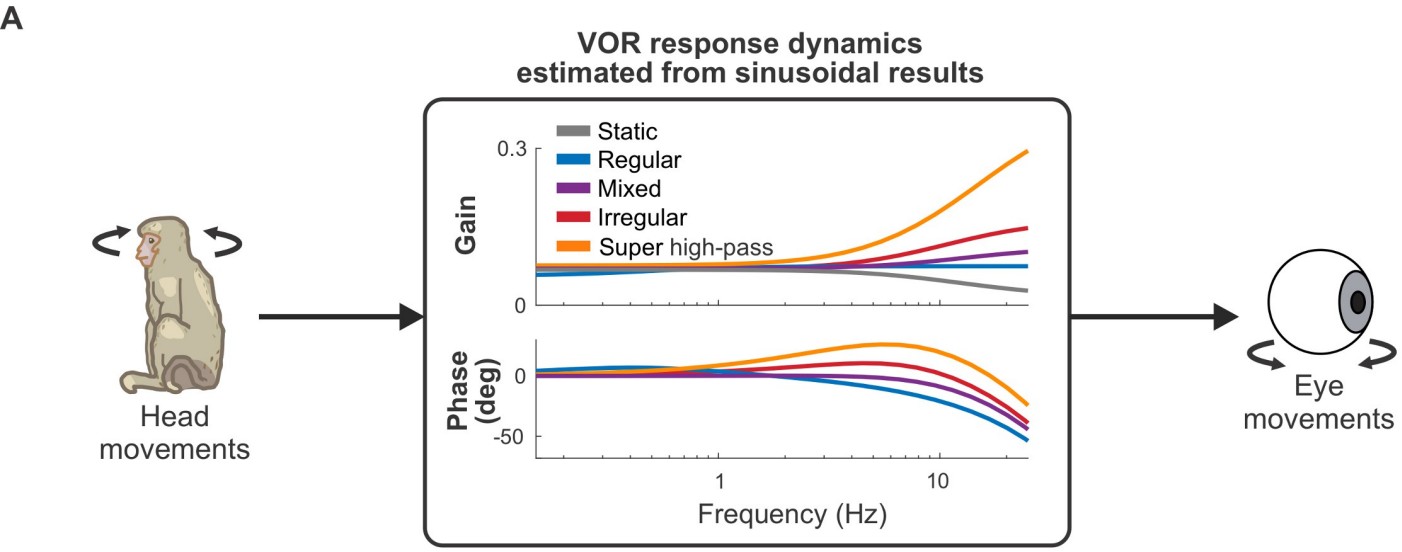

**B**

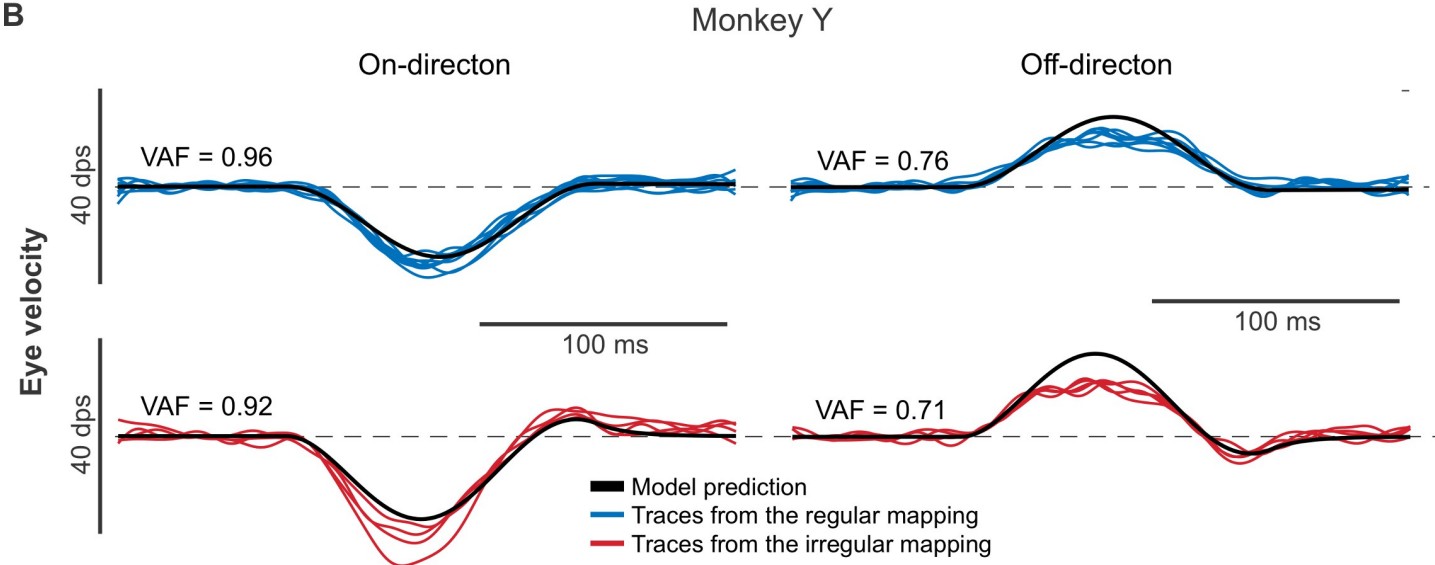

**Fig 5. Eye movement responses to virtual transient stimulation can be predicted by those to sinusoidal stimulation.** (**A**) The schematic of the model that estimates the VOR response dynamics of each monkey using a simple linear transfer function (see Materials and methods). (**B**) The model prediction (black line) of the evoked eye movements during transient head movements in the on (left column) and off (right column) directions for the regular (top row) and the irregular (bottom row) mappings.

Materials and methods) to the input–output relationship between sinusoidal prosthetic stimulation and eye movements recorded in our study above (Fig 3). We then used this model to predict the transient VOR responses described above. Fig 5B shows example model fits with the regular (top) and irregular (bottom) mappings for Monkey Y. We found that a model based on VOR eye movement responses to sinusoidal prosthetic stimulation well-predicted VOR responses to transient prosthetic stimulation (mean VAF = 0.91 and 0.90 in the on-direction for Monkeys Y and G, respectively; see S8 Fig for all VAF values). Notably, the model tended to slightly underpredict the evoked VOR in the on-direction (eye movements away

from the side of the implant; Fig 5B, left) while overpredicting the evoked VOR in the off-direction (eye movements toward the side of the implant; Fig 5B right). This asymmetry results from unilateral stimulation and is similar to that observed in animals with unilateral vestibular deficits (see, for example, [16,31]).

## Conventional rate mappings artificially limit the prosthesis performance

To account for the fact that the firing rates of afferents saturate at 350 to 400 spikes/s [11,32], prior applications of vestibular prostheses in patients [15,20] and nonhuman primates [16,33] have generally programmed a static sigmoidal nonlinearity into the mapping of rotational head velocity to pulse rate. Here, we programmed a similar sigmoidal nonlinearity with a slightly higher saturation rate (500 pps) into our prosthesis mappings (see Materials and methods; also, Figs 1B and 6A). We speculated that this programmed nonlinearity was responsible for the reduction in VOR gain (saturation) observed above for our super high-pass 2× gain mapping at 20 Hz (Fig 3E, red arrows). To confirm this proposal, we plotted the post-sigmoid (linear–nonlinear cascade) pulse rate versus the pre-sigmoid (linear only) pulse rate. As expected, this particular mapping was the only one to deviate from the unity line (S9A Fig, red arrow). Thus, the programmed static sigmoidal nonlinearity could account for, at least partially, the saturation observed with this mapping.

Because we set the maximum pulse rate higher than the normal saturation range, namely to 500 pps, we next investigated whether the ability to reach such high post-sigmoid pulse rate resulted in afferent saturation and further contributed to the VOR saturation. To do this, we pooled the experimental data across all mappings and plotted the maximum evoked eye velocity as a function of the maximum stimulation rate for each sinusoidal frequency, as well as for the transient condition (Fig 6B). Surprisingly, we found that the observed relationship was approximately linear across the entire stimulation range, suggesting that our mappings did not result in afferent saturation (R squared = 0.99 and 0.91 for Monkey G and 0.96 and 0.79 for Monkey Y, respectively, for 20 Hz and transient stimulations, both of which had the stimulation extended into the 400 to 500 pps range; yellow shaded areas of Fig 6B).

At first glance, this observed linearity was unexpected given that afferents saturate at 350 to 400 sp/s. However, the conventional inclusion of a static sigmoidal nonlinearity explicitly mimicking afferent saturation inherently assumes that the efficacy between electrical stimulation and vestibular afferent firing is 1:1 (i.e., 100%). Based on our experimental findings, we speculated that the actual stimulation efficacy was significantly lower than 1:1. To test this proposal, we asked whether we could explain the relationship between the stimulation rate and the evoked eye movements using a standard control systems-based model of the VOR (see Materials and methods; Fig 6C, blue box) with stimulation efficacy <100% (Fig 6C, red box). Indeed, our model best predicted the observed eye velocity when the value of the stimulation efficacy was markedly less than 1:1 (i.e., 28% and 4.5% for Monkeys G and Y, respectively; Fig 6D). Notably, our prediction of approximately 28% efficacy in Monkey G corresponded well to the approximately 27% efficacy showed by a prior study in which single afferents were directly recorded during stimulation with the same prosthesis design [18]. Furthermore, it is likely that the prosthesis stimulation induced a decrease in the synaptic efficacy of afferent-vestibular nuclei neuron transmission, as was also established in this same prior study. Accordingly, to account for this effect, we included 17% central pathway depression in the VOR model (S9B Fig). Simulation of this adjusted model yielded slightly higher estimates of stimulation efficacy (34% and 5% for Monkeys G and Y, respectively; S9C Fig). If our approximately 28% to 34% stimulation efficacy estimate is correct, then our maximum stimulation rate of 500 pps would have translated into only approximately 140 to 170 sp/s of afferent firing (Fig

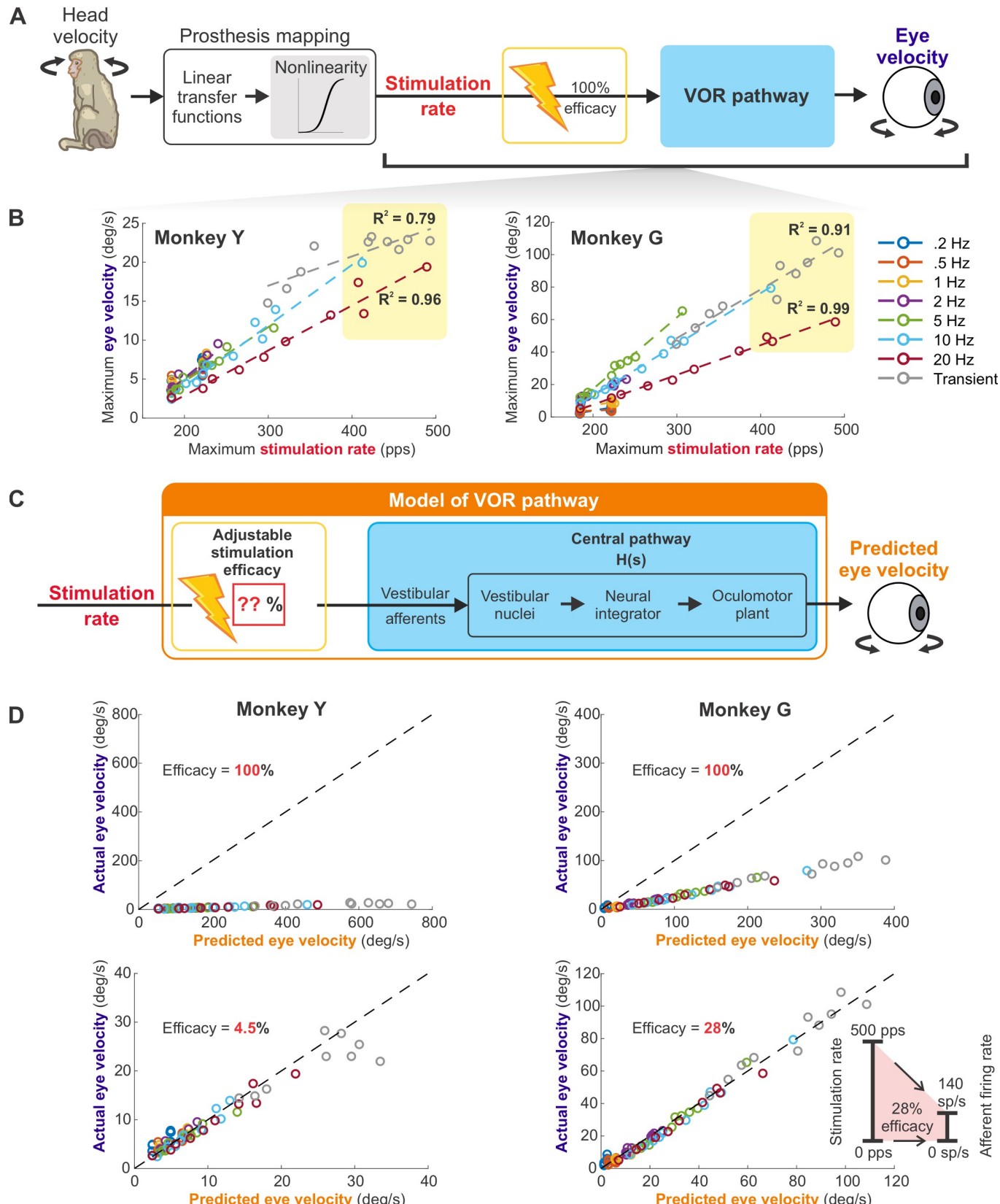

**Fig 6. Conventional approach in programming the static nonlinearity artificially constrained the dynamic range of prostheses to an unnaturally narrow band.** (**A**) Schematic of the VOR generation in a monkey implanted with the prosthesis. (**B**) The maximum evoked eye velocity plotted as a function of the maximum stimulation rate. The relationship remained linear even at the 400–500 pps stimulation range, as indicated by the yellow shaded areas. (**C**) The schematic of the model that could predict the results in (**B**). The model combines the adjustable stimulation efficacy with the control systems-based model of the VOR pathway. In addition, this model accounted for the low-pass behavior of the oculomotor premotor pathway (blue box) responsible for the decrease in slope observed with increasing modulation frequency in (**B**). (**D**) Plots of the actual eye velocity and the predicted eye velocity from the model in (**C**) using 100% stimulation efficacy (top) and the lower stimulation efficacy that best fit the data (bottom). The dashed line is the unity line. (Inset) Schematic showing that 500 pps maximum stimulation rate only translated to 140 sp/s afferent firing rate at 28% efficacy. Data underlying this figure can be found at https://doi.org/10.5281/zenodo.6338639.

6D, inset), which could explain the observed linearity. We hypothesize that the even lower efficacy estimated for Monkey Y was likely due to less optimal electrode placement. Taken together, our results suggest that prior studies that programmed the maximum stimulation rate to be exactly the afferent saturation rate utilized only a fraction of the available firing rate range and therefore artificially constrained the dynamic range of prostheses to an unnaturally narrow band.

## Discussion

### Implementing natural encoding strategies optimizes vestibular prosthesis performance

Our central finding is that by applying our understanding of the brain's natural coding strategies, we can improve vestibular prosthesis performance. Specifically, incorporating the natural dynamics of vestibular afferents into the mapping of head motion to pulsatile stimulation produced more temporally accurate VOR eye movements. Further exploration of the parameter space revealed that the mapping incorporating more extreme high-pass tuning than is naturally displayed by vestibular afferents produces an unnatural phase advance, whereas the static mapping without dynamics produces VOR eye movements that substantially lag stimulation. Trends were comparable for virtual and actual head rotations, with gains enhanced in the latter case. Using computational methods, we further demonstrate that the same model could explain the eye movements evoked by sinusoidal and transient prosthetic stimulation and that a stimulation efficacy substantially less than 100% could account for our results. Taken together, our results underscore the benefits of leveraging knowledge of endogenous afferent dynamics and activation efficacy to ensure vestibular sensorimotor accuracy, which could directly translate to better functional outcomes in patients.

### Neurophysiologically derived mappings better compensate for neural pathways delays

During everyday activities, the vestibular input experienced by humans and monkeys has significant power up to 20 Hz [28,29]. Over this frequency range, vestibular afferents show strong high-pass tuning that produces a 2-fold increase in response modulation at 20 versus 0.5 Hz. Further, while afferents modulate in phase with rotational head velocity at the lower end of this frequency range (<0.5 Hz), they also demonstrate increasing phase leads as a function of frequency [10–12]. In this context, it is interesting that while the tuning of vestibular afferents has been well characterized, it had not been incorporated in design of vestibular prosthetic devices. Notably, mappings used currently in clinical trials directly map instantaneous angular head velocity to a specific pulse rate—equivalent to the static mapping in our present study (e.g., [15,20,21]). Likewise, static mappings have largely been used in prior studies in primate animal models [16,34–36]. As expected, our static mapping generated VOR responses consistent with the results of these prior studies, which were characterized by poor phase

compensation. Finally, it is noteworthy that prior work by Merfeld and colleagues employed a dynamic mapping between angular head velocity to a specific pulse rate [33,37–39]. However, this mapping only approximated afferent dynamics over a very low frequency range (<0.1 Hz) and thus performed in a comparable manner to the static mapping over most of the natural frequency range (i.e., 0.1 Hz to 20 Hz).

In contrast to the poor phase compensation displayed by both the static mapping in this study and other prior studies, our biomimetic mappings exhibited much more natural phase behavior. In healthy primates, the VOR is compensatory across the natural frequency range with eye velocity nearly in phase with head velocity, as well as gains close to 1 even at 20 Hz [22,23]. The prevailing view is that this compensatory performance is achieved because the high-pass tuning of vestibular afferents account for the finite approximately 5 ms delay of VOR pathway (i.e., synaptic, neural, and muscle activation times from afferents to extraocular muscles), as well as the response dynamics of the VOR pathway [22,40]. Our present results using virtual stimulation provide direct support for this view and emphasize the critical role of neural response dynamics in ensuring accurate sensorimotor transformations. A further advantage of our virtual stimulation condition was that it isolated the vestibular system's contribution to the VOR. Interestingly, we likewise found similar trends across mappings when the same patterned prosthetic stimulation was delivered during actual physical rotation. Importantly, VOR response gains were enhanced in this latter condition, such that the gain increase was more than the sum of the gains during physical rotation with no stimulation or just stimulation alone. We speculate that this enhanced response was due to the presence of additional congruent extravestibular signals (i.e., proprioception, tactile, and visual) augmenting neuronal responses to modulated prosthetic stimulation and/or a heightened arousal state during the actual motion stimulation.

## Dual coding strategies support 2 functional streams

Central neurons in the vestibular nuclei that generate the VOR receive direct monosynaptic inputs from both types of vestibular afferents, but inputs from regular afferents predominate [24–27]. Here, we correspondingly found that mappings that mimicked regular afferent dynamics or a mix of regular/irregular afferents dynamics evoked VOR eye movements with temporal properties that best matched the encoded head motion stimulus (Fig 4A). In contrast, as noted above, our static mappings did not compensate for VOR neural circuitry delay and/or oculomotor plant dynamics, resulting in large phase lags. Moreover, the VOR eye movements evoked by our more dynamic irregular and super high-pass mappings displayed unnatural advances in phase and timing relative to the applied sinusoidal and transient stimulation, respectively.

In this context, there is evidence that the relative contributions of regular versus irregular afferents to the central pathways mediating the vestibulo-spinal reflex versus the VOR are optimized to match the dynamic requirement of each pathway [25,26,41]. As noted above, regular afferent dynamics are well suited to offset the VOR pathway delays and the biomechanics of the oculomotor plant. On the other hand, the larger gains and phase leads of the irregular afferents are required to drive robust postural responses since vestibulo-spinal pathways must account for the higher inertia of the body relative to the eye [27,42]. Furthermore, irregular afferents likely also make important contributions to ascending pathways that convey vestibular information to cortical brain circuits mediating self-motion perception ([43]; also reviewed in [14]). Interestingly, a recent imaging study in humans reported that these cortical circuits overlap with those mediating postural control [44]. Thus, we speculate that the biomimetic irregular mapping might be better suited for postural control and self-motion perception, whereas the biomimetic regular and mixed mappings produced the most temporally accurate VOR eye movements as was observed in our present study.

## The trade-off between response amplitude and coding range also constrains neuroprostheses

The timing characteristics of the VOR evoked by our regular and mixed mappings produced temporally accurate responses, yet the VOR evoked by these mappings was not fully compensatory (i.e., gain less than 1). To better understand why this was the case, we probed the parameter space in 2 different dimensions. First, we tested whether implementing a mapping with dynamics that were even more high pass than those of irregular afferents improved gain performance. This "super high-pass" mapping produced relatively higher VOR gains that in 1 monkey actually approached unity at the highest frequencies (S2 Fig). However, it is important to note that due to the steepness of the gain curve, this was not the case at lower frequencies (compare 0.2 to 1 Hz and 2 to 20 Hz; S2 Fig). Second, we tested how implementing a higher mapping gain overall affects the evoked VOR. In general, we found that doubling the gain of any of our mappings produced roughly 2-fold larger VOR eye movement responses with similar phase behavior as the original mappings (Fig 3E). However, we also found that for our most dynamic mapping (i.e., super high-pass), the gain of the evoked VOR saturated and then began to decrease starting at approximately 10 Hz.

Overall, the above analysis of the parameter space highlights the trade-off problem inherent in mapping head motion to stimulation rate. Improvements in VOR gain can be accomplished by increasing the overall mapping gain, yet this will decrease the dynamic range that the prothesis can encode before cutoff and saturation. This trade-off problem is further exacerbated by artificial constraints from the common approach in programming the static nonlinearity used in prior studies, both in humans and animal models [15,16,20,33]. Setting the maximum stimulation rate to be exactly the afferent saturation rate inherently assumes that the relationship between prosthetic stimulation and afferent firing is 1:1 (100% stimulation efficacy). However, our present modeling results suggest that the efficacy was actually substantially less, consistent with our prior experimental results [18]. In this prior study, the experimentally measured afferent stimulation efficacy was only approximately 27%, for monkey with comparable VOR eye movement gain as Monkey G (see Fig 3 of [18])—a value similar to that estimated here (28%). We thus speculate that the generally low prosthesis-evoked VOR gains reported in the literature (human participants [15,21]; monkeys [16,34]) are, at least in part, due to artificially limiting the dynamic range of stimulation relative to the firing rate of the afferents. We propose that the use of mappings, which actually account for the efficacy of afferent activation (Fig 6D), would lead to improved VOR gains, while maintaining the dynamic range required to represent the natural range of head movements experienced in everyday life [28,29,45]. In parallel to the implementation of such mappings, innovative methods based on direct current stimulation could become a potential alternative and/or complementary approach for increasing stimulation efficacy [46–48].

Finally, we note that at least 2 additional factors likely combine to limit prothesis performance. First, our prior single unit recording studies in the vestibular nuclei have shown that prosthetic pulsatile stimulation rapidly reduces the efficacy of the afferent-central neuron synapse, which, in turn, reduces the gain of the prosthesis-evoked VOR eye movement [18]. As discussed in this previous report, the pulsatile stimulation produced by the prosthesis appears to induce long-term depression at this synapse due to the synchrony evoked across the vestibular afferent population—such synchrony is not present during natural head motion. Second, following bilateral loss, it is an open question how the central pathways adapt to process vestibular signals, because the vestibular contribution cannot be measured in the animals without a prosthesis. On one hand, it is possible that central mechanisms could increase the synaptic weighting of the afferent input as has been observed following unilateral peripheral loss

[49,50]. On the other hand, it is possible that, in the case of bilateral peripheral loss, adaptive central mechanisms instead function to suppress the contribution from any residual vestibular afferent nerve input. Future experiments focused on distinguishing between these 2 possibilities can serve to inform the development of improved prothesis-based approaches to restore vestibular function.

## Vestibular implant users will likely benefit from incorporating natural encoding strategies

Vestibular prostheses are the only restorative treatment for patients with bilateral vestibular deficits who cannot compensate centrally via rehabilitation exercises. However, the benefits of vestibular implantation must be weighed against the cost and risks of surgery [20,51] in an analysis that depends directly on the prosthesis functional performance. Here, we found that biomimetic mappings resulted in the same trends in functional improvement in both monkeys. However, we also observed substantially lower gains in 1 monkey. Comparable variability in evoked VOR gains has likewise been reported across subjects in prior human [15,21] and monkey [16,34] studies. One factor that could contribute to such gain variability is differences in the precise placement of the electrodes targeting the ampullae [52]. Indeed, optimization of electrode placement is a focus of ongoing research [53,54]. In this context, our finding that biomimetic mappings enhance performance despite gain differences across subjects directly advocates for incorporating such coding strategies in clinical practice. Furthermore, the implementation of such mappings does not increase cost or risk from additional surgery or hardware changes.

## Advances in vestibular prosthesis can inform the development of other sensory prostheses

Our present findings serve to extend a growing body of literature showing improved functional performance from biomimetic sensory prosthesis designs (e.g., enhanced speech intelligibility for cochlear implants [5]; improved manual dexterity for limb prostheses that include tactile feedback [55–57]). Notably, our present findings show that accounting for the specific afferent input to a well-defined sensorimotor pathway can optimize functional performance (i.e., VOR). As discussed above, VOR pathways receive inputs from 2 afferent classes (regular and irregular), with the input from the former being stronger (reviewed in [14]). Correspondingly, our stimulation mappings that best matched this afferent input produced the most temporally accurate VOR. In contrast, we postulate mappings that predominately account for the dynamics of irregular afferents may be best suited to restore other vestibular functions including posture and self-motion perception. More generally, studies in other sensory systems have likewise shown heterogeneity in the afferent inputs mediating different behavioral and perceptual outcomes. For example, in the somatosensory system, slow adapting mechanoreceptors detect sustained force and are thought to better aid in adjusting grip force, while fast adapting receptors detect changes in force and are thought to contribute to the perception of vibration (reviewed in [58]). We suggest that the development of prostheses that can account for such pathway specific heterogeneities will be essential to improving functional outcomes across sensory systems.

## Materials and methods

### Study design

The goal of this study was to evaluate whether biomimetic coding strategies, when implemented in the mapping between head motion and stimulation rate, improve prosthesis

performance, as measured by the VOR accuracy. Testing was done in 2 macaque monkeys with bilateral vestibular loss and a vestibular prosthesis comparable to that currently implanted in human patients in an ongoing clinical trial [15,20]. We first compared the VOR evoked by 2 mappings mimicking the natural response dynamics of the 2 types of vestibular afferents to that evoked by the standard conventional mapping currently used clinically. Specifically, semi-circular canal afferents demonstrate "high-pass dynamics" characterized by an increase in gain and phase lead as a function of head motion frequency (see Fig 1B, blue and red), which contrasts with the flat gain and lack of phase lead characterizing the conventional mapping (see Fig 1B, gray). Additionally, we explored the parameter space by testing mappings with dynamics between those of these 2 afferent types as well as a mapping with an even greater level of high-pass tuning than vestibular afferents. Data analyses were done in a blind manner, i.e., without knowing which mapping was used for each dataset. The number of repetitions for each paradigm and exclusion criteria are described in detail under "Data analysis" below.

## Surgical procedures

Two macaque monkeys (Monkey Y, male, 8 kg; Monkey G, female, 9 kg) were used for all experimental paradigms. All housing, surgical, and experimental protocols were approved by the Johns Hopkins Animal Care and Use Committee. Animals were housed on a 12-h light/dark cycle with daily enrichment. Throughout the study, animals were monitored in consultation with the clinical veterinarian staff to ensure physical and emotional well-being. Euthanasia procedure follows the best practices recommended by the clinical veterinarian staff and the ACUC. Both monkeys were initially fitted with a head implant for head fixation and scleral coils for eye position recording (detailed methods in [16,18]). In brief, isoflurane was used to maintain surgical level of anesthesia (2% to 3% initially then 0.8% to 1.5% throughout the procedure). A stainless steel post was affixed to the skull with stainless steel screws and dental acrylic. A 15- to 16-mm coil, made of 3 turns of Teflon-coated stainless steel wire, was sutured to the scleral beneath the conjunctiva of the eye.

Both monkeys were also implanted with vestibular prothesis electrodes targeting each ampulla (detailed methods in [16]; right ear for Monkey Y and left ear for Monkey G). Monkey Y was implanted with its canals intact, while Monkey G had received gentamicin treatment via bilateral intratympanic injection. A mastoidectomy was performed under sterile conditions to allow access to the labyrinth. Two small holes were made at the junction of the ampullae of the superior and horizontal semicircular canals and a forked electrode array was inserted to target the 2 canals. A hole was also made in the thin segment of the posterior semicircular canal near its junction with the ampulla and a single-tine electrode array was inserted. Additionally, reference electrodes were inserted into the common crus of the labyrinth and in extracranial musculature. To stabilize the electrodes, pieces of fascia and bone were inserted around each array. The electrode leads were run under the periosteum and secured to the headcap with dental acrylic. The animals recovered for 2 weeks before any experiments were performed.

Each monkey had bilateral vestibular loss confirmed by VOR gain <0.1 at 2 Hz in both directions of rotation. This was because vestibular implantation disrupted the canal function, causing vestibular hypofunction in the implanted side. In addition, both monkeys also had deficits in their contralateral ear: Monkey Y had nonfunctioning old prosthesis electrodes in the left ear and Monkey G had bilateral gentamicin treatment via intratympanic injection and, subsequently, a labyrinthectomy in the right ear due to insufficient reduction of the VOR gain contributed by the right ear. Current experiments were done 10 years after implantation for Monkey G and 2 years after implantation for Monkey Y. Both monkeys were use in other studies with the same implant. Stimulation was only delivered during experiments and not in home cages.

## Mapping angular head velocity into stimulation rates

The vestibular prosthesis (adapted from [16]) consisted of a gyroscope module that senses angular rotation, a processing module that converts head angular velocity into stimulation rate according to a head-velocity-to-pulse-rate mapping and sends a trigger to an external current source (AM Systems), which delivers a charge-balanced biphasic pulse (200 μS/phase, cathodic first) to the canal electrodes. The stimulation current was set at the maximum level for each monkey: 170 μA for Monkey G (80% of the facial threshold) and 250 μA for Monkey Y (the safe charge limit based on the electrode size). In this study, we modified the mapping function to consist of a cascade between a linear transfer function and a fixed cutoff nonlinearity, which has been shown to accurately describe vestibular afferent behaviors [45].

For the linear part, we assumed that the linear firing rate r(t) as a function of head movement velocity stimulus $V_H(t)$ is given by: $r(t) = (H * V_H)(t) + r0$, where the asterisk denotes convolution and r0 denotes the baseline firing rate (chosen to be 150 Hz). The Fourier transform of H(t), i.e., the transfer function, is in the form:

$$H(f) = k \frac{S(S + \frac{1}{T_1})}{(S + \frac{1}{T_C})(S + \frac{1}{T_2})}, \tag{Eq 1}$$

where $S = i2\pi f$. In particular, we used constants similar to those used in [45] to estimate the transfer functions of the regular and irregular afferents. For regular afferents: k = 5.056 pps/dps, $T_1$ = 0.0175 s, $T_2$ = 0.0027 s, and $T_c$ = 5.7 s. For irregular afferents: k = 38.889 pps/dps, $T_1$ = 0.03 s, $T_2$ = 0.0006 s, and $T_c$ = 5.7 s. In addition, we introduced 2 new transfer functions. One is called the "super high-pass" transfer function, which has a higher level of high-pass behavior than irregular afferents in nature. The transfer function is in the form given above but with k = 76.76 and $T_1$ = 0.06, which were doubled their normal values for irregular afferents. The other is called the "mixed" transfer function, which behaves like a mix between regular and irregular afferents. The transfer function is given by:

$$H_{mixed}(f) = n(H_{regular} + H_{irregular})/2, \tag{Eq 2}$$

where n is a constant (1.5923) used to match the gain at 0.5 Hz to that used by [16] (i.e., 0.78 pps/dps) without changing the phase relationship. The static mapping has no high-pass response dynamics with a flat gain of 0.78 pps/dps and zero phase lead.

For the fixed cutoff nonlinearity, a sigmoidal function was used similar to [16,45] and is given by the form:

$$r_{final}(t) = c_3/(1 + e^{-c_1(r_{linear}(t) - c_2)}), \tag{Eq 3}$$

where $r_{linear}$ is the linear firing rate from the transfer functions and $r_{final}$ is the actual final firing rate. Other constants were chosen such that the upper limit of the firing rate was 500 pps and the gain in the middle region (centered around the baseline firing rate of 150 pps) of the sigmoid is 1: $c_1 = 4/c_3$, $c_2 = 150 + \log(c_3/150 - 1)/c_1$, and $c_3 = 500$.

For the double mapping gain conditions, we multiplied each gain constant k by 2, while keeping other constants the same for the regular, irregular, and super high-pass mappings. For the mixed mapping, we multiplied n (in Eq 2) by 2 while keeping other constants the same: n = 3.1846. For the static mapping, the flat gain was doubled to 1.56 pps/dps.

## Data acquisition

Monkeys sat comfortably in a primate chair with head fixation in darkness while prosthesis pulses were delivered. Scleral search coil technique was used to measure horizontal and vertical

eye position. Eye data were digitized at 1 kHz (Blackrock Microsystems). Trigger pulses indicating the start of stimulation paradigms were concurrently digitized at 30 kHz and later synchronized with the eye data. Room light was turned on between stimulation paradigms to maintain alertness and reduce darkness visual adaptation. Juice rewards were intermittently given to maintain alertness.

## Stimulation conditions

For the "virtual head motion" condition, the prosthetic pulses were delivered as if the monkeys' heads were moving, but the monkeys were actually stationary in the chair. For the "physical head motion" condition, comparable prosthetic pulses were also delivered but evoked by actual physical head motion that was detected by the gyroscope module of the prosthesis during applied whole-body rotation. Two types of stimuli were used: Sinusoidal motion. Sinusoidal rotations with 50 deg/s at 7 frequencies spanning the naturalistic range (0.2, 0.5, 1, 2, 5, 10, 20 Hz) were used. Each presentation consisted of 10 sinusoidal cycles and at least 2 presentations were recorded for each experimental condition. Due to motion platform system limitations, we only delivered the actual motion for the "physical head motion" condition up to 10 Hz. Transient motion. Brief, transient motions (around 150 ms long and 200 deg/s peak velocity) in both directions were used to mimic natural head turns. These velocity profiles were then converted to the stimulation pulse rate via the different mappings as described above (see section "**Mapping angular head velocity into stimulation rates**"). At least 20 presentations were recorded for each experimental condition.

## Data analysis

All analyses were performed using MATLAB 2019b (Mathworks). Digitized (1 kHz) eye position data were low-passed filtered at 125 Hz (51st degree, Hamming window, filtfilt function in MATLAB). Eye velocity data were then calculated by differentiating the eye position data. Saccades were detected using thresholding and removed from the dataset. Slow-phase eye velocity data were then further low-pass filtered at 50 Hz (51st degree, Hamming window, filtfilt function in MATLAB).

**Sinusoidal data.** A dynamic linear regression technique was used to find the gain, offset, and the shift (phase) of the entire cycle of VOR response. For 0.2 and 0.5 Hz, only cycles with at least 40% slow phase were included and at least 3 cycles were included in the analysis. For higher frequencies, only cycles with 60% slow phase were included and at least 10 cycles were included in the analysis. The log-linear slope of the gains at high frequencies (5 to 20 Hz) was estimated using linear regression after transforming the frequencies to the logarithmic scale. The gain ratio was calculated by dividing each ×2 gain data point by the mean ×1 gain. The standard error of change in phase was nonparametrically estimated using bootstrap methods with 2,000 iterations. VOR responses from the "physical head motion" condition were quantified from difference between the mapped stimulation condition and baseline only condition to compute the component of the VOR evoked by the prosthesis.

**Transient data.** Only trials with no saccade 50 ms prior and after the stimulus were included. At least 4 trials were included in the analysis. The response latency was estimated by fitting a linear fit to the 10 data points before and after the eye movements crossed the 2 SD thresholds. The intersection between the fitted line and the x-axis was then used as the start of the response. The gain of the response was calculated by dividing the maximum eye velocity by the amplitude of the head movement. The peak timing is the time point where the eye movement reached its peak velocity. Transient VOR responses from the "physical head motion" condition were quantified as described above for sinusoidal data.

## Modeling

**Input–output modeling.** A linear transfer function approximating the gain and phase response was obtained for each monkey using the invfreqs function in Matlab. To account for the fixed pathway delay, the phase response was shifted back by 6 ms before the transfer function estimation and the simulated output of the model was then shifted forward 6 ms. The transfer function with 1 zero and 1 pole worked well for Monkey Y and the transfer function with 1 zero and 2 poles worked well for Monkey G. The model fit was assessed using variance accounted for (VAF):

$$VAF = 1 - \frac{Var(y - \hat{y})}{Var(y)}, \tag{Eq 4}$$

where $y$ is the actual data and $\hat{y}$ is the model estimation.

**Pathway modeling.** Overall, a linear systems–based VOR pathway model [1,59] was used to predict the eye velocity from the afferent firing rate:

$$EV(f) = T_{VN}(f)T_{NI}(f)T_{Plant}(f)AFR(f) \tag{Eq 5}$$

where $EV(f)$ is the Fourier transform of the eye velocity, $AFR(f)$ is the Fourier transform of the afferent firing rate from the earlier part of the model. The other terms are the transfer functions of different parts of the VOR pathway:

$$\text{Vestibular nuclei } T_{VN}(f) = -g_{VOR}\frac{T_{VOR}}{(sT_{VOR}+1)}\frac{(sT_c+1)}{T_c} \tag{Eq 6}$$

$$\text{Neural integrator } T_{NI}(f) = T_{e1} + \frac{1}{s} \tag{Eq 7}$$

$$\text{Plant } T_{Plant}(f) = \frac{se^{-s\tau}}{(sT_{e1}+1)(sT_{e2}+1)} \tag{Eq 8}$$

where $s = 2\pi i f$, $i = \sqrt{-1}$. $g_{VOR}$ is the VOR gain, which was set to yield perfect compensation at 2 Hz rotation in normal monkeys. $T_c$ = 5.7 $s$ is the time constant of canal afferent. $T_{VOR}$ = 16 $s$ is the VOR time constant. $\tau$ = 0.006 $s$ is the pathway delay. These constants were taken from previous literature [1]. $T_{e1}$ and $T_{e2}$ are time constants describing the neural integrator and plant dynamics; $T_{e1}$ cancels out when Eqs 7 and 8 are combined, whereas $T_{e2}$ was set to 0.008 and 0.025 for Monkey Y and Monkey G, respectively, to account for the low-pass dynamics of their oculomotor plants (i.e., see different slopes in Fig 6B).

To compute the input afferent firing rate to this model for each rotation movement condition/prosthesis mapping, the prosthesis pulse rate was generated using the linear nonlinear cascade as described above. To account for the high-pass behavior seen in the prosthesis-evoked VOR response at low frequencies (Fig 2B, gray; see also [15,16,34]), an extra high-pass filter was applied to the afferent firing rate. For each animal, the filters were estimated from the gray traces in Fig 2B (cutoff frequency of 3.5 Hz and 0.2 Hz for Monkey G and Monkey Y, respectively). In our modeling of VOR eye movements, we first assumed a 1:1 relationship between pulse rate and afferent firing rate (i.e., setting the stimulation efficacy to 100%), which overestimated the eye velocity. We next adjusted the stimulation efficacy to estimate the best fit to our data across conditions in each animal. Finally, to account for the 17% reduction in prosthesis-evoked eye movements that occurs due to the reduction of central VOR pathway efficacy [18], we created another model that incorporated this 17% reduction.

## Statistical analysis

All data were presented as mean ± SEM. MATLAB 2019b was used to conduct statistical analysis. Normality assumption of each dataset was assessed using Lilliefors test prior to other statistical analyses. For normally distributed datasets, two-tailed *t* test was used. For all other datasets, nonparametric Wilcoxon signed rank test was used. Statistical significance was determined at $p < 0.05$ for all tests.

## Supporting information

**S1 Fig. VOR data of each monkey prior to normalization.** (**A**) VOR gains prior to normalization across natural frequency range (0.2–20 Hz). Note that the figure shows the same data as Fig 2B but the gains are displayed on the absolute scale (not normalized). (**B**) Example VOR traces prior to normalization. Dashed lines indicate inverted, virtual head velocity. The shaded areas indicate SEM. Red, blue, and gray refer to the irregular, regular, and static mappings, respectively. Data underlying this figure can be found at https://doi.org/10.5281/zenodo.6338639.
(TIF)

**S2 Fig. VOR data of each monkey prior to normalization with 2 additional mappings.** (**A**) VOR gains prior to normalization across natural frequency range (0.2–20 Hz). Note that the figure shows the same data as Fig 3C but the gains are displayed on the absolute scale (not normalized). (**B**) Example VOR traces prior to normalization. Dashed lines indicate the inverted, virtual head velocity. The shaded areas indicate SEM. Yellow, red, purple, blue, and gray refer to the super high-pass, irregular, mixed, regular, and static mappings, respectively. Data underlying this figure can be found at https://doi.org/10.5281/zenodo.6338639.
(TIF)

**S3 Fig. VOR gain and phase plots for the X2 condition.** (**A**) Normalized VOR gains across natural frequency range (0.2–20 Hz), using the same normalization reference as Fig 2. Note the reduction in gain at 20 Hz due to saturation. (**B**) The phase response of the VOR. Note the similarity to the phase response in the X1 condition in Fig 3D. The shaded area indicated the SEM. Yellow, red, purple, blue, and gray refer to the super high-pass, irregular, mixed, regular, and static mappings, respectively. Data underlying this figure can be found at https://doi.org/10.5281/zenodo.6338639.
(TIF)

**S4 Fig. VOR data during physical sinusoidal rotation in the dark.** (**A**) VOR gains across the frequency range (0.2–10 Hz). (**B**) The phase response of the VOR. (**C**) Example VOR traces. Dashed lines indicate inverted head velocity. There was a significant increase in the VOR gain mainly over the 5–10 Hz range for Monkey G ($p < 0.05$, Bonferroni corrected) but not for Monkey Y. There was also a significant increase in phase lead mainly over 5–10 Hz range for both monkeys ($p < 0.05$, Bonferroni corrected). There was no significant difference in the gain and phase responses for early and late cycles in both monkeys. Yellow, red, purple, blue, and gray refer to the super high-pass, irregular, mixed, regular, and static mappings, respectively. Data underlying this figure can be found at https://doi.org/10.5281/zenodo.6338639.
(TIF)

**S5 Fig. VOR data during physical sinusoidal rotation in the light.** (**A**) VOR gains across the frequency range (0.2–10 Hz). (**B**) The phase response of the VOR. (**C**) Example VOR traces. Dashed line indicated the inverted head velocity. There was a significant increase in the VOR gain mainly over the 5–10 Hz range for Monkey G ($p < 0.05$, Bonferroni corrected) and at 10 Hz for Monkey Y ($p < 0.05$, Bonferroni corrected). There was also a significant increase in

phase lead mainly over 5–10 Hz range for both monkeys ($p < 0.05$, Bonferroni corrected). There was no significant difference in the gain and phase responses for early and late cycles in both monkeys. Yellow, red, purple, blue, and gray refer to the super high-pass, irregular, mixed, regular, and static mappings, respectively. Data underlying this figure can be found at https://doi.org/10.5281/zenodo.6338639.
(TIF)

**S6 Fig. Evoked VOR during virtual transient head movements in the off-direction.** (**A**) Traces of the evoked eye movements for Monkey G during transient head movements (off-direction), normalized as in Fig 4. Dashed lines indicate the inverted head velocity. Dotted vertical line indicates the start of the head movements. Solid vertical line denotes the peak of the head movements. Short colored vertical lines indicated the estimated onset of the evoked eye movement. (**B**, **C**, and **D**) Quantification of the traces in (A) for normalized gain [using the same normalization reference as in Fig 4], time difference between eye and head velocity peaks, and onset latency, respectively. Results from Monkey G and Monkey Y are plotted on the left and on the right, respectively. Error bars indicate the SEM. Yellow, red, purple, blue, and gray refer to the super high-pass, irregular, mixed, regular, and static mappings, respectively. Similar to the on-direction results, the biomimetic regular and mixed mappings evoked a peak VOR response that was well aligned with the peak head velocity (not significantly different from 0 ms except for Monkey G mixed mapping, which showed a significant but small lead of 12.3 ± 1.7 ms, $p < 0.01$, Bonferroni corrected). In contrast, the static mapping resulted in the VOR peak with a significant delay ($p < 0.001$ for Monkey G, not significant for Monkey Y, Bonferroni corrected), while the VOR peaks evoked by irregular and super high-pass mappings actually led the stimulus ($p < 0.05$ for Monkey G, not significant for Monkey Y). Data underlying this figure can be found at https://doi.org/10.5281/zenodo.6338639.
(TIF)

**S7 Fig. Evoked VOR during physical transient head movements in the on-direction.** (**A**) Traces of the evoked eye movements for Monkey G during physical transient head movements (on-direction), normalized as in Fig 4. Dashed lines indicate the inverted head velocity, scaled to facilitate timing comparison. Dotted vertical line indicates the start of the head movements. Solid vertical line denotes the peak of the head movements. Short vertical lines indicated the estimated onset of the evoked eye movement. (**B**, **C**, and **D**) Quantification of the traces in (**A**) for normalized gain, time difference between eye and head velocity peaks, and onset latency, respectively. Results from Monkey G and Monkey Y are plotted on the left and on the right, respectively. Error bars indicate the SEM. Yellow, red, purple, blue, and gray refer to the super high-pass, irregular, mixed, regular, and static mappings, respectively. Data underlying this figure can be found at https://doi.org/10.5281/zenodo.6338639.
(TIF)

**S8 Fig. VAFs of the model prediction of virtual transient eye velocity using the virtual sinusoidal data.** The VAFs are plotted for each trial of the head movements in the on (top row) and off (bottom rows) directions for Monkey Y (left column) and Monkey G (right column). Error bars indicate the SEM. Data underlying this figure can be found at https://doi.org/10.5281/zenodo.6338639.
(TIF)

**S9 Fig. Conventional approach in programming the static nonlinearity artificially constrained the dynamic range of prostheses to an unnaturally narrow band.** (**A**) Post-sigmoid stimulation rate plotted as a function of pre-sigmoid stimulation rate. The red arrow points to

the data point from the super high-pass 2× gain mapping at 20 Hz, which showed large deviation from linearity. (**B**) Similar to the model in Fig 6C but with the addition of the CNS adaptation (17%; [18]). (**C**) Plots of the actual eye velocity and the predicted eye velocity from the model in (**B**) using 100% stimulation efficacy (top) and the lower stimulation efficacy that best fit the data (bottom). The dashed line is the unity line. Data underlying this figure can be found at https://doi.org/10.5281/zenodo.6338639.
(TIF)

## Acknowledgments

We thank V. Chang, M. Christman, O. Leavitt, R. Mildren, T. Niebur, O. Stanley, E. Tourney, R. Wei, and O. Zobeiri for critically reading the manuscript.

## Author Contributions

**Conceptualization:** Kantapon Pum Wiboonsaksakul, Kathleen E. Cullen.

**Formal analysis:** Kantapon Pum Wiboonsaksakul.

**Funding acquisition:** Charles C. Della Santina, Kathleen E. Cullen.

**Investigation:** Kantapon Pum Wiboonsaksakul.

**Methodology:** Kantapon Pum Wiboonsaksakul, Dale C. Roberts, Charles C. Della Santina, Kathleen E. Cullen.

**Supervision:** Charles C. Della Santina, Kathleen E. Cullen.

**Writing – original draft:** Kantapon Pum Wiboonsaksakul, Kathleen E. Cullen.

**Writing – review & editing:** Kantapon Pum Wiboonsaksakul, Dale C. Roberts, Charles C. Della Santina, Kathleen E. Cullen.

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
