## [Editor Report · Decision Letter 0]

4 Mar 2022

Dear Dr Cullen, 

Thank you for submitting your manuscript entitled "Restoring vestibular afferent dynamics enhances sensorimotor performance: implications for vestibular prostheses" for consideration as a Research Article by PLOS Biology.

Your manuscript has now been evaluated by the PLOS Biology editorial staff, as well as by an academic editor with relevant expertise, and I am writing to let you know that we would like to send your submission out for external peer review.

Once your full submission is complete, your paper will undergo a series of checks in preparation for peer review. Once your manuscript has passed the checks it will be sent out for review. To provide the metadata for your submission, please Login to Editorial Manager (https://www.editorialmanager.com/pbiology) within two working days, i.e. by Mar 06 2022 11:59PM.

If your manuscript has been previously reviewed at another journal, PLOS Biology is willing to work with those reviews in order to avoid re-starting the process. Submission of the previous reviews is entirely optional and our ability to use them effectively will depend on the willingness of the previous journal to confirm the content of the reports and share the reviewer identities. Please note that we reserve the right to invite additional reviewers if we consider that additional/independent reviewers are needed, although we aim to avoid this as far as possible. In our experience, working with previous reviews does save time. 

If you would like to send previous reviewer reports to us, please email me at ggasque@plos.org to let me know, including the name of the previous journal and the manuscript ID the study was given, as well as attaching a point-by-point response to reviewers that details how you have or plan to address the reviewers' concerns. 

Given the disruptions resulting from the ongoing COVID-19 pandemic, please expect some delays in the editorial process. We apologise in advance for any inconvenience caused and will do our best to minimize impact as far as possible.

Kind regards,

Gabriel

Gabriel Gasque

Senior Editor

PLOS Biology

ggasque@plos.org

---

## [Decision Letter · Decision Letter 1]

15 Apr 2022

Dear Dr Cullen,

Thank you for submitting your manuscript "Restoring vestibular afferent dynamics enhances sensorimotor performance: implications for vestibular prostheses" for consideration as a Research Article at PLOS Biology. Your manuscript has been evaluated by the PLOS Biology editors, an Academic Editor with relevant expertise, and by several independent reviewers.

In light of the reviews (below), we will not be able to accept the current version of the manuscript, but we would welcome re-submission of a much-revised version that takes into account all of the reviewers' comments. In considering the reviewer feedback, we particularly note that all of the Reviewers expressed concerns with the low VOR-evoked gain you observe and Reviewer 1 raised some additional concerns with your virtual head motion design. The Academic Editor agreed that a particular concern is the low gain of the prothesis based VOR and that this is further complicated by the gain difference you see between the 2 animals. They also suggested that a potential explanation for the low gains, might, in fact, be related to the study reliance on direct stimulation rather than stimulation reflecting natural head movements. 

In any revision we would therefore be particularly looking to see how you address the concerns related to gain levels and head motion. Along these lines, if possible, we ask that you add date on prosthetic stimulation patterns reflecting natural head movements. If you do not have such data and the experimental animals are no longer available for further assessment, we would be looking for a convincing argument as to why a consideration of natural head movements is not needed to appreciate that relevance of "nature-like" stimulation patterns. This is particularly important given the potential usage of your findings for prostheses design and would be a consideration for any final decision on your study. Additionally, we would expect to see a substantial revamping of the study presentation in any revision to PLOS Biology to ensure your study is accessible to our broad scientific readership. This should also include some expansion to the discussion to further touch on the broader implications of your work beyond the vestibular system and how this fits in with evidence from other domains – something you only touch on very briefly at the moment.

Please note that we cannot make any decision about publication until we have seen the revised manuscript and your response to the reviewers' comments. Your revised manuscript is also likely to be sent for further evaluation by the reviewers.

We expect to receive your revised manuscript within 3 months. 

**IMPORTANT - SUBMITTING YOUR REVISION**

*Re-submission Checklist*

*Published Peer Review*

*PLOS Data Policy*

*Blot and Gel Data Policy*

Sincerely,

Kris

Kris Dickson

Neurosciences Senior Editor/Section Manager

PLOS Biology

kdickson@plos.org

REVIEWS:

Reviewer's Responses to Questions

PLOS authors have the option to publish the peer review history of their article (what does this mean?). If published, this will include your full peer review and any attached files.

Reviewer #1: No

Reviewer #2: No

Reviewer #3: No

Reviewer #1: This is an extremely interesting and provocative paper. It is well and clearly written and the figures are superb. That is this manuscript certainly deserves to be published. I have only one major remark concerning the choice of recording the VOR in head fixed monkeys. The rest of my comments, apart from typos, concerned the "pedagogic" aspects of the introduction and result sections. This paper deserves a large audience and particularly it should be understandable by ENT doctors, being its topic. However, the fact is that some paragraphs, and concepts, which are highlighted below, may be difficult to understand by readers (including the ENT) not too familiar with the vestibular physiology and its physiopathology. These passages should be corrected to ameliorate their comprehension.

More precisely:

I would suggest a more detailed title: Restoring vestibular afferent dynamics in Macaque monkeys enhances sensorimotor performance: implications for vestibular prostheses in Human

Introduction

The authors state that their protocols used "virtual head motion "in the two monkeys with complete vestibular loss. That is, the prosthesis pulses were delivered as if the monkeys' heads were moving, but they were actually stationary in the chair. Why this option was chosen? It means that the afferent information encoded by the spindles of the neck muscles did not occur during these virtual head motion, which could have impacted the VOR since proprioceptive information converge with the vestibular information in the vestibular nuclei and reticular formation. That would plead to test the stimulations during real head movements. Why it was not done? Please comment that point for the reader. 

The vestibular system detects our head motion relative to space. In turn, this information is used to generate essential stabilizing reflexes and to provide us with our subjective sense of motion and orientation.

I would suggest a more detailed sentence to help the uninformed reader: . In turn, this information is used to generate essential stabilizing reflexes and complex motor synergies to control gaze and posture and to provide us with our subjective sense of motion and orientation.

First, the three axes of head rotation are encoded by three separate sensory organs (i.e., the semicircular canals),

I understand that sentence, but again it may be confusing for the uninformed reader not to mention the otoliths sensors in one way or the other at that stage.

To address this question, here we directly tested whether implementing the natural response

dynamics of vestibular afferents improves prosthesis performance.

Please add that this was done in a primate model 

parameter space

Please define that concept

Using computational methods, we then demonstrated that the same model could account for the eye movements evoked by both sinusoidal and transient stimulation, and that the efficacy between stimulation pulses and afferent firing is substantially less than 1:1 as often assumed.

This point is important and the description is too sketchy to be easily understood by an average reader at that stage of the manuscript. If I get it well, do the authors mean that it is often assume that a single pulse of stimulation would trigger each time one (or several?) action potentials? If it is the case how a computational model can test that hypothesis and why an experiment approach was not used also.

Methods

by two mappings mimicking the natural high-pass response dynamics of the two types of vestibular afferents

A more detailed explanation of the terms "natural high-pass response dynamics" is required for the reader unfamiliar with the vestibular neurophysiology and possibly a figure to illustrate that point also.

All surgical and experimental protocols were approved by the Johns Hopkins Animal Care and Use Committee. 

Date of approval, reference of the demand and as stated by PNAS

Animal Research (involving vertebrate

animals, embryos or tissues)

Provide the name of the Institutional Animal

Care and Use Committee (IACUC) or other

relevant ethics board that reviewed the

study protocol, and indicate whether they

approved this research or granted a formal

waiver of ethical approval

*

Include an approval number if one was

obtained

*

If the study involved non-human primates,

add additional details about animal welfare

and steps taken to ameliorate suffering

The vestibular function, the way it is written for the two monkey is confusing and need to be revised. Monkey Y was implanted with canal intact in the right ear. In addition, it had non-functioning old prosthesis electrodes in the left ear. Reading that description, this monkey should have an intact vestibular function. Why it ended up with a model of bilateral vestibular loss? Monkey G had received gentamicin treatment via bilateral intratympanic injection. Presumably this one should have a bilateral vestibular loss as a result of the injection. In that condition, why subsequently, a labyrinthectomy in the right ear was performed and why on one side only?

Typo problems in my text : 170 uA and 250 uA. Presumably micro Amp ?

digitized at 30 kHz and later synced: what mean syned? Synchronized

Reviewer #2: In this study, the authors investigate how to optimize vestibular prostheses (that restore the sense of rotation by stimulating the vestibular organs in the inner ear). They demonstrate that incorporating in the prosthesis a transfer function that mimics the vestibular afferents' dynamics improves the temporal accuracy of behavioural responses in two rhesus monkeys. This study is an important step in the development of neuroprotheses, and also an impressive technical achievement.

The main conclusion of the study (regarding the dynamics of the response) is solidly supported. 

However, one limitation of vestibular prostheses is that their response gains are very low, and it is the case in this study. The authors propose that this is due to a poor stimulation efficiency between the prosthesis and the afferent, but this conclusion isn't be well supported, and should be reconsidered.

Major remarks:

The author observe that the gain of the VOR evoked by the prosthesis is 28% in one animal and 4.5% in the other. To explain this, they propose that the stimulation efficiency (i.e. the frequency of the discharge of the vestibular afferent relative to the pulse rate of the prosthesis) is 28% or 4.5% respectively. This can explain the observed data, indeed. However, the authors don't consider any alternative hypothesis. One possible alternative is that central VOR pathways readapt following the vestibular lesion. Another alternative is that the prosthesis evokes synchronous firing that is less efficient at driving VOR. None of these hypothesis can be discounted based on the author's data. Therefore, the authors haven't demonstrated that stimulation efficiency is less than 1:1, and this conclusion should be removed from the abstract, introduction, etc.

Note that this point has major implications for the development of vestibular prosthesis. If the low gain of the VOR is indeed attributable to a low stimulation efficiency, then the implication is that developing better stimulation methods could eventually restore normal VOR. On the other hand, if it is due to central adaptation (including possibly physiological changes in the vestibular nuclei), then this adaptation is the bottleneck. 

Figure 6 and S6: the gain of the response is much higher in animal G (28%) compared to Y (4.5%). The author show animal G in the main figure 6 and Y in the supplementary figure S2: this is not a very balanced way to present data. It feels like the authors emphasize animal G and, in layman language, 'hide the bad monkey' in the supplementary figure. Given how the figure is organized, there is all the space necessary to show both animals in Fig. 6, and no justification not to do so. 

Figure 2A: superimposing the 'normalize eye velocity' to an 'inverted head velocity' is incorrect. First, the head is not actually moving so it would be proper to refer to a 'simulated' head velocity. Second, head velocity has a unit of degrees per second, whereas the normalized eye velocity is unitless. Third, and fundamentally, the authors are giving the impression that the VOR is approximately compensatory at 10Hz, which is not the case (Fig. S1). Merging Fig. S1 and Fig. 2 would be a more balanced way to present the results.

Minor remarks:

Figure 3: when comparing Fig. 3A and Fig. 3C, it appears that the normalized gain of the VOR resembles a lot the gain of the afferent's transfer functions in Fig. 3A. In other words, in seems that the gain of the VOR compared to stimulation amplitude is constant. Given this, wouldn't it be sensible to program a transfer function with a flat gain, which would presumably induce a VOR with a constant gain along frequencies? 

Methods: could you give more details about the animal's history? How much time elapsed between the implantations and the recordings shown here? Was the prosthetics continuously active in their home cage (or used on a regular basis)? Were they used in other studies? With the same implants?

Reviewer #3: This is a excellent piece of scientific work in a challenging area. I have no major criticisms.

The absolute gain responses are relatively low as shown in the supplementary information. I think at least an insert of the absolute gains should be shown in one of the main figures to give an idea of the current state of play of vestibular prostheses.

Did the investigators at any point, give real inertial stimuli in the light with the vestibular prosthesis turned on? If the vestibular prosthesis is only ever switched on in the dark to generate virtual eye movements, then there is no opportunity for the system to calibrate (and adapt) itself to the artificial input, irrespective of how well the investigators attempt to emulate natural stimuli. 

Hence, have the authors assesed sinusoidal VOR in the light with prothesis on and off? If so, does the VOR gain improve over time?

In this regard, the authors mention that "the vestibular system is uniquely suited for prosthesis development" with which I agree. But there is little mention in the introduction about the development of the prosthesis being a clinically useful device for patients. In that regard, the authors should mention that human patients generally adapt well to bilateral vestibular loss with exercise so the benefits of a vestibular prosthesis on top of vestibular exercises would need to be balanced by the risks of the procedure/device. This risk-benefit curve would tilt one way or the other depending upon the relative efficacy and safety of the prosthesis. Such an argument would be useful in explaining the need for the current animal-based work (to improve efficacy and lower risks).

Has there been any finite element modelling to assess how the e-fields are affected by changing the stimulus parameters, which in turn may influence the impact upon local and not so local neural structures. I am not saying this needs to be done, simply it would be interesting to know if this has been done.

The assumptions and coefficients on pg 14 could only be easily understood by researchers who performed the studies that were referenced, but would otherwise require me to dig into the original papers. Is there anyway to justify the various parameter selections a little more without over-inflating the manuscript?

Page 16 - "Pathway modeling. Overall, a linear systems based VOR pathway model (Glasauer, 2007;Robinson, 1981) was used to predict the eye velocity from the afferent firing rate:" Does this statement assume that the afferent firing rate is 1:1 with stimulus frequency (as presumably you are not measuring the afferent firing rate - see next comment).

But, I think an important finding is the implication of a 28% (or 4.5% for monkey Y) stimulus efficacy. If this is an important barrior for clinical implementation, can the authors comment about the potential to surmount this barrier? 

For the reported P values, can the mean and the statistic be consistently reported. Also, when relevant, correct the P value for multiple comparisons.

---

## [Decision Letter · Decision Letter 2]

2 Aug 2022

Dear Dr Cullen,

Thank you for your patience while we considered your revised manuscript "Restoring vestibular afferent dynamics enhances sensorimotor performance in Macaque monkeys: implications for vestibular prosthesis use in humans" for publication as a Research Article at PLOS Biology. This revised version of your manuscript has been evaluated by the PLOS Biology editors, the Academic Editor and the original reviewers.

Based on the reviews and feedback from our Academic Editor, we are likely to accept this manuscript for publication, provided you satisfactorily address the minor discussion comments from Reviewer 3, and the data and other policy-related requests at the bottom of this email.

Additionally, in order to ensure your study is as accessible as possible to our broad audience, we would like to suggest a slight title change to this work. We'd suggest: "A prosthesis utilizing natural vestibular encoding strategies improves sensorimotor performance in monkeys".

As you address the various journal-related items, please take this last chance to review your reference list to ensure that it is complete and correct. If you have cited papers that have been retracted, please include the rationale for doing so in the manuscript text, or remove these references and replace them with relevant current references. Any changes to the reference list should be mentioned in the cover letter that accompanies your revised manuscript.

We expect to receive your revised manuscript within two weeks. 

*Published Peer Review History*

*Press*

Sincerely,

Kris

Kris Dickson, Ph.D. (she/her)

Neurosciences Senior Editor/Section Manager,

kdickson@plos.org,

PLOS Biology

ETHICS STATEMENT:

Thank you for the information provided on the work you've performed in NHPs. When resubmitting the final version, please double check that this information fully complies with our policies here:

https://journals.plos.org/plosbiology/s/animal-research#loc-non-human-primates

In particular, non-human primate studies must be performed in accordance with the recommendations of the Weatherall report “The use of non-human primates in research”. Manuscripts describing research involving non-human primates must include details of animal welfare, including information about housing, feeding, and environmental enrichment, and steps taken to minimize suffering, including use of anesthesia and method of sacrifice if appropriate.

DATA POLICY:

Regardless of the method selected, please ensure that you provide the individual numerical values that underlie the summary data displayed in the following figure graphs/heat maps as they are essential for readers to assess your analysis and to reproduce it: Fig2, Fig3, Fig4, Fig6BD, Supplemental Figures (all): 1-9

(Also note that main Fig 4 currently has panel C above panel B which is a bit confusing)

Please also ensure that FIGURE LEGENDS in your manuscript include information on where the underlying data can be found, and ensure your supplemental data file/s has a legend. (This step is often forgotten!!!)

DATA NOT SHOWN?

- Please note that per journal policy, we do not allow the mention of "data not shown", "personal communication", "manuscript in preparation" or other references to data that is not publicly available or contained within this manuscript. Please double check your submission and either remove mention of any such data or provide figures presenting the results and the data underlying the figure(s).

Reviewer remarks:

Reviewer #1: The reviewer thanks the authors to have taken into account all his suggestions and in particular to have include new data on prosthetic stimulation patterns evoked by actual physical head movements. 

Reviewer #2: The authors did a very nice job at revising an already impressive manuscript. I particularly appreciate the thoughtfulness with which they responded to my remarks, and I have no further comments to make.

Reviewer #3: I am satisfied with the author's replies and changes in the manuscript in response to my comments. The changes further strengthen the paper.

I have two small comments about the Discussion.

The paragraph in Discussion Ln 447 is important as it highlights how different stimulus settings may suite different vestibular mechanisms, e.g. VOR vs. perception vs. postural control, important for future optimisation of the prosthesis - as they note in Ln 456 - "Thus, we speculate that the biomimetic irregular mapping might be better suited for postural control and self-motion perception, whereas the biomimetic regular and mixed mappings produced the most temporally accurate VOR eye movements as was observed in our present study.", hence....

- the authors should make a note in their manuscript that in humans, cortical networks mediating vestibular perception and postural control actually overlap in the right temporal cortex in the inferior longitudinal fasciculus (Calzolari et al., Brain DOI: 10.1093/brain/awaa386). It follows that in line with their comments, future prosthesis optimisation should involve perceptual and postural measures.

- In Ln455 they should qualify the contribution of ascending pathways to perception a little more precisely. These pathways specifically convey the vestibular signals to cortical brain circuits and it is these brain circuits that are more closely engaged in the processes that mediate vestibular perception (add reference doi: Hadi et al. BioRxiv https://doi.org/10.1101/2021.12.03.471139). This seems a little pedantic since what the authors wrote was not wrong, but the suggested change seems clearer to my mind.

---

## [Editor Report · Decision Letter 3]

16 Aug 2022

Dear Dr Cullen,

Thank you for the submission of your revised Research Article "A prosthesis utilizing natural vestibular encoding strategies improves sensorimotor performance in monkeys" for publication in PLOS Biology. On behalf of my colleagues and the Academic Editor, Peter Thier, I am pleased to say that we can in principle accept your manuscript for publication, provided you address any remaining formatting and reporting issues. These will be detailed in an email you should receive within 2-3 business days from our colleagues in the journal operations team; no action is required from you until then. Please note that we will not be able to formally accept your manuscript and schedule it for publication until you have completed any requested changes. 

When you are making any of these requested changes, we'd also encourage you to upload a spreadsheet-based form of the underlying data for your study (e.g Excel-based). While the MATLAB files you've deposited in Zenodo will be helpful to researchers in the field, not all of our interested readership will necessarily have the ability to open and access the .mat file format. 

PRESS

We frequently collaborate with press offices. If your institution or institutions have a press office, please notify them about your upcoming paper at this point, to enable them to help maximize its impact. If the press office is planning to promote your findings, we would be grateful if they could coordinate with biologypress@plos.org. If you have previously opted in to the early version process, we ask that you notify us immediately of any press plans so that we may opt out on your behalf.

Sincerely, 

Kris

Kris Dickson, Ph.D. (she/her)

Neurosciences Senior Editor/Section Manager

PLOS Biology

kdickson@plos.org